# Study on the Solder Joint Reliability of New Diamond Chip Resistors for Power Devices

Wenyu Wu [1], Geng Li [2], Shang Wang [2,*], Yiping Wang [2], Jiayun Feng [2], Xiaowei Sun [1] and Yanhong Tian [2,*]

[1] NO. 38 Research Institute, China Electronics Technology Group Corporation, Hefei 230031, China; hzhcandy@126.com (W.W.)
[2] State Key Laboratory of Advanced Welding and Joining, Harbin Institute of Technology, Harbin 150001, China; li-geng@hit.edu.cn (G.L.)
* Correspondence: wangshang@hit.edu.cn (S.W.); tianyh@hit.edu.cn (Y.T.)

**Abstract:** New diamond chip resistors have been used in high-power devices widely due to excellent heat dissipation and high-frequency performance. However, systematic research about their solder joint reliability is rare. In this paper, a related study was conducted by combining methods between numerical analysis and laboratory reliability tests. In detail, the shape simulation and thermal cycling finite element simulation for solder joints with different volumes were carried out. The optimized solder volume was 0.05 mm$^3$, and the maximum thermal cycling stress under the optimized shape was 38.9 MPa. In addition, the thermal cycling tests with current and high temperature storage tests were carried out for the optimized solder joint, which showed good agreement with the simulation results, clarified the growth and evolution law of intermetallic compound at the interconnection interface, and proved the optimized solder joint had great anti-electromigration, temperature cycling and high temperature storage reliability. In this work, an optimized solder joint structure of a diamond chip resistor with high reliability was finally obtained, as well as providing considerable reliability data for the new type of diamond chip resistors, which would boost the development of power devices.

**Keywords:** diamond chip resistor; solder joint; reliability; numerical analysis





## 1. Introduction

High-power electronic devices are widely used in the field of wireless communication [1,2]. With the demand for high performance and light weight of electronic devices, the continuous improvement of their integration has brought a significant increase in power density and the large current and high heat have led to many reliability problems [3]. Numerous studies have shown that most of the failures of electronic devices come from the interconnection solder joints of electronic packaging due to the stress caused by the thermal mismatch of materials and the evolution of microstructure and morphology [4]. As the most common component of electronic devices, chip resistors are used in almost every electronic device. Therefore, ensuring the reliability of chip resistor solder joints under high temperature and power cycling is the cornerstone of the development of high-power devices.

With continuous power and frequency improvement of power devices, the commonly used materials Al$_2$O$_3$ of chip resistor substrate cannot meet the requirements of heat dissipation [5,6]. The thermal conductivity of CVD diamond is 2000 W·m$^{-1}$·K$^{-1}$, which is dozens of times higher than Al$_2$O$_3$. Using diamond as the substrate material of chip resistors can greatly improve the heat dissipation performance [6]. In addition, this new type of diamond chip resistor has advantages of high frequency (more than 30 GHz), high power, small size, light weight and stable performance, and can service in extreme environments such as deep space exploration and military equipment that require ultra-high reliability [7]. The usage of new materials always leads to new reliability problems. There have been

many studies on the reliability of traditional chip resistor solder joints [8,9], but the research on the reliability of new chip resistor solder joints is not sufficient. Therefore, it is of great significance to supplement the reliability data of the new diamond chip resistors to ensure the stable operation of power devices and assist the development of power devices.

Due to the frequent change of ambient temperature or switch of power devices, chip resistors need to undergo temperature cycling. Under the alternating temperature load, the thermal mismatches of different interfaces of materials generate large thermal stress, which leads to plastic deformation, grain boundary slip and grain boundary defects in solder joints. Stress concentrates at those defects and results in initial cracks, then cracks propagate to cause failure [10]. It is necessary to study the reliability of new chip resistors under temperature cycling. The finite element analysis method can save a lot of time and cost in the reliability analysis process, which can quickly obtain the stress and strain response of the device under a certain load and predict the service life of solder joints by combining a constitutive equation and life prediction model [11–15]. The accuracy of the finite element model greatly affects the accuracy of simulation and the simulation of the solder joint shape plays an important role in improving the accuracy of model. In addition, the shape of the solder joint directly determines the stress distribution of the solder joint. Therefore, the solder joint shape with the lowest stress level can be obtained through the iterative optimization of solder joint shape simulation and thermal cycle finite element simulation.

The new diamond chip resistors in power devices need to work at a high temperature due to high power. The continuous high temperature causes excessive growth of brittle intermetallic compounds (IMCs) at the soldering interface, which leads to an increasing risk of failure. The mechanism of IMCs growth involves the diffusion and migration of elements. The diffusion of elements is affected by many factors such as element concentration, temperature, stress, etc. [16], therefore, it is difficult to evaluate the reliability of solder joints only by simulation, so the high temperature storage (HTS) experiment is essential.

In order to realize the electrical connection of the diamond chip resistors, it is necessary to electroplate electrodes at the terminals of the resistor. The electrode material is usually nickel. The quality of the nickel coating will significantly affect the reliability of the solder joint in the process of temperature cycling or aging [17]. Therefore, this paper also focuses on the analysis of the reliability of the coating interface.

With the background mentioned above, this study conducted the solder joints shape simulation and thermal cycling finite element simulation for the new diamond chip resistors. In order to verify the simulation results, corresponding thermal cycling tests were carried out. The power devices also bore current load during service, so another control group with current was set in order to evaluate the impact of current on solder joints' reliability. In addition, high temperature storage tests were carried out to evaluate the reliability of the solder joints of the new diamond chip resistors stored under high temperature for a long time.

## 2. Materials and Methods

### 2.1. Numerical Simulation

In order to investigate the device-level reliability, the power divider with new diamond chip resistors was selected as the study object. Several assumptions were made to ensure the accuracy and feasibility of the numerical simulation as listed below:

- The metal plating was ignored;
- All materials were uniform and dense;
- All interconnecting interfaces were tightly combined;
- The effect of gravity was considered in the simulation of solder joint shape;
- The change of material thermodynamic parameters with temperature was considered in the thermal cycling simulation.

The process of numerical simulation was shown in Figure 1. Parameterized modeling was carried out in the Surface Evolver software (version 2.70) according to the size data of the chip resistor. The differential equation of surface wetting was set to calculate the solder joint

shape under different solder volumes by the minimum energy principle. Then, the Surface Evolver (SE) model of the chip resistor was imported into ANSYS (version 14.0), and the parametric modeling of the whole power divider was carried out. The Anand model was used to describe the viscoplastic behavior of solder under a low stress state. The thermodynamic simulation used thermal element SOLID70 and structural element SOLID185. Tetrahedral elements were used for meshing due to plenty of irregular structures, and the mesh of the solder part was refined to improve the simulation accuracy. Fixed constraints were applied at the bolt holes at the four corners of the power divider shell, as shown in Figure 2a. The thermal cycling profile was in accordance with the requirements of TC4 in IPC9701. The temperature was from −55 to 125 °C, one cycle for 50 min, whose rising time and falling time were 15 min. Some research showed the inelastic strain amplitude of the solder joint was generally stable after 7 to 8 cycles [11–13], so a total of 10 thermal cycles were applied to the model, as shown in Figure 2b. The material parameters used in the numerical simulation can be found in Tables S1 and S2 of the Supplementary Materials.

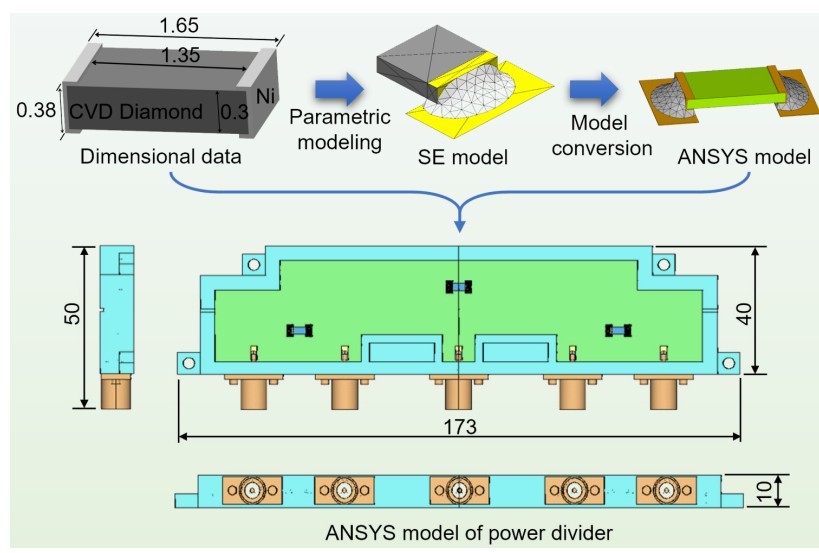

**Figure 1.** Schematic process of numerical simulation.

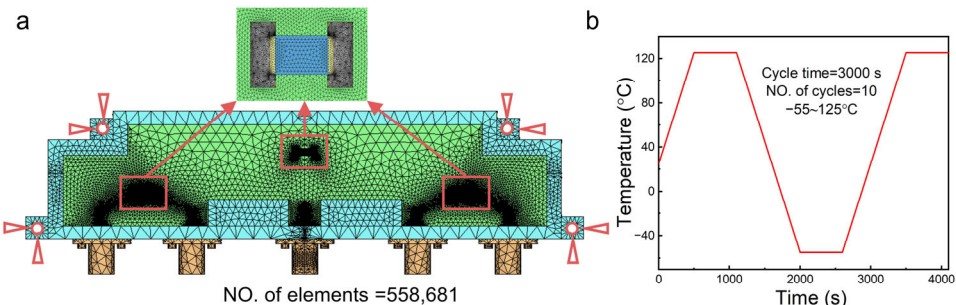

**Figure 2.** Pretreatment of finite element simulation: (**a**) Mesh generation and constraints; (**b**) Load files.

### 2.2. Laboratory Experiment

Screen printing and reflow soldering process were carried out according to the optimal solder volume obtained by simulation, and the new diamond chip resistors were soldered onto the microstrip plate of power divider. The chip resistors were the CRD0603DX5W2 model produced by Smiths Interconnect (London, UK) and the solder was eutectic SnPb. The corresponding thermal cycling tests were carried out on the power divider by high-low temperature test-box to verify the simulation results, and the samples after 200, 500, and 1000 cycles were taken out for analysis. In order to determine the anti-electric migration ability of the solder joint, another energized control group was

set with the current of 0.15 A. The aging temperature of HTS test was set to 150 °C by constant temperature drying box, and the temperature accuracy was ±0.1 °C. The growth rate of intermetallic compounds followed Fick's diffusion law and was proportional to the quadratic power of time, so aging times of the samples were set to be 1 day, 4 days, 9 days, 16 days, 25 days, 36 days, and 49 days.

The obvious hardness difference made it difficult to obtain the flat interconnection interface between the diamond and SnPb solder, so the interface was polished with a cross-section polisher. The direction of the ion beam was perpendicular to the plane of the microstrip plate and polished 6 h for each sample. The microstructure of the solder joints was investigated by the scanning electron microscopy (SEM, ZEISS Sigma 300, Jena, Germany), and the SEM images were taken by backscattered electrons (BE) to obtain compositional contrast. The composition of IMCs were identified by energy dispersive spectrometer (EDS, ZEISS Sigma 300), and the thickness measurement of interface IMCs was calculated by the Image J (version 1.8.0) according to the imaging contrast difference of different materials and the thickness was an average value of all samples under same conditions. We took three chip resistors for each test condition to avoid contingency.

## 3. Results and Discussion

The simulation results of solder joint shape are shown in Figure 3, where V is the volume of solder. The solder was only at the bottom of the resistor with the volume of 0.01 mm$^3$, and the lateral wall of the resistor was not wet. As the volume of solder increased, the solder began to wet the lateral wall of the resistor and continued to climb. When the volume came to 0.025 mm$^3$, solder climbed more than half the height of the resistor. The solder climbed to the top of the resistor when the volume became 0.05 mm$^3$, but the width direction of the resistor was not completely wetted and the solder joint presented a fillet shape of chip components as usual at this time. When the volume of the solder increased further, the solder continued to wet on the pad and the direction of resistor width until the volume of solder reached 0.13 mm$^3$, the interfacial tension between solder and pad could not resist the gravity of the solder at this time, so the solder collapsed and overflowed the pad area. According to the SMT general test standard IPC-610F, the solder of chip components should be higher than 1/2 of the component's height, so we took the solder joints with the volume of 0.025 mm$^3$, 0.05 mm$^3$, 0.075 mm$^3$, 0.01 mm$^3$, and 0.025 mm$^3$ for subsequent thermal cycling simulation.

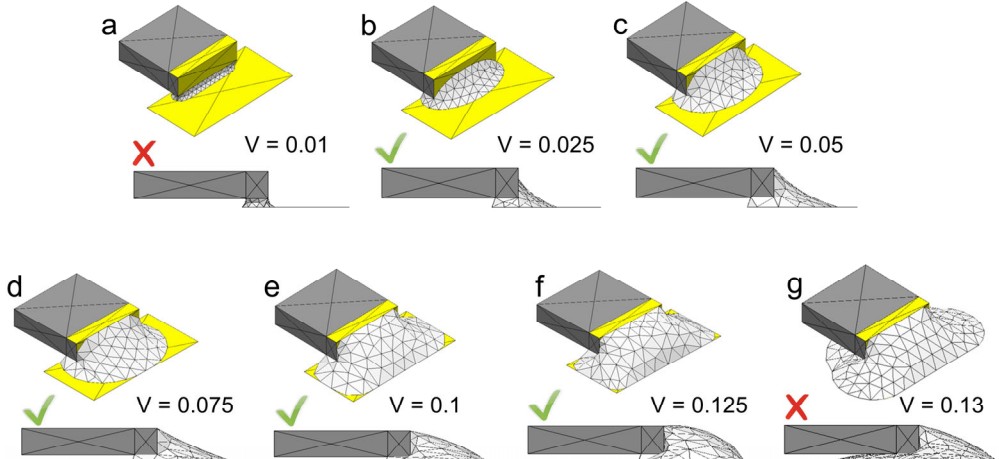

**Figure 3.** Simulation results of solder joint shape under the different solder volumes of (**a**) 0.01 mm$^3$, (**b**) 0.025 mm$^3$, (**c**) 0.05 mm$^3$, (**d**) 0.075 mm$^3$, (**e**) 0.1 mm$^3$, (**f**) 0.125 mm$^3$, (**g**) 0.13 mm$^3$.

The thermal cycling simulation was taken for solder joints with different solder volumes in power divider with chip resistors. Among the three chip resistors carried by the power divider, the thermal stress of the solder joints near four-corner screw holes was the

highest because they were close to the fixed constraint and had difficulty releasing the stress through deformation. During a whole thermal cycle, stress of the joints increased to the highest when the temperature just fell to minimum, because the CTE of each material had a larger difference in the low temperature. The stress distribution of the solder joint near the screw hole at the lowest temperature is shown in Figure 4. The maximum von Mises stress and strain were produced at the corner where the inner side of the solder joint contacted with the lower edge of the resistor. Because the horizontal distance between the inner side of solder pad and the resistor was very small, the solder joint there showed a nearly right angle and the stress was concentrated at this part. The maximum stress of solder joints of different shapes was between 39.0 and 43.4 MPa, and there was no significant difference. Although the maximum stress was slightly higher than the tensile strength of SnPb solder (36 MPa), the high stress here was only concentrated in a very short time and a very small area, after which the stress could be released through microcracks here, which did not affect the overall reliability of solder joints.

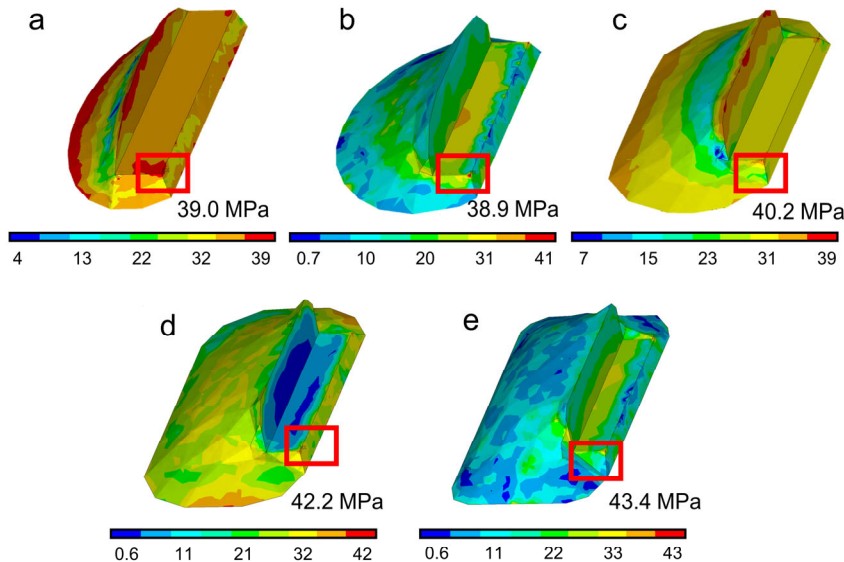

**Figure 4.** Stress distribution of solder joints at the lowest temperature with the solder volume of (**a**) 0.025 mm$^3$, (**b**) 0.05 mm$^3$, (**c**) 0.075 mm$^3$, (**d**) 0.1 mm$^3$, (**e**) 0.125 mm$^3$.

In Figure 4a, the solder joint was too small to release the stress through deformation of the solder joint, resulting in the highest-level overall stress when the solder volume was 0.025 mm$^3$. The stress level of solder joints with solder volume of 0.075 and 0.01 mm$^3$ was also large, which was due to the terrible shape of solder joints. The overall stress of solder joints with solder volume of 0.05 and 0.0125 mm$^3$ was much lower, but too much solder may lead to a Manhattan effect of the chip resistor during reflow soldering. Through the above analysis, the solder joint with a volume of 0.05 mm$^3$ was considered to be the best.

The stress-strain curve at the critical point of the solder joint with a volume of 0.05 mm$^3$ is shown in Figure 5. The curve was in the shape of a gradually convergent hysteresis loop, and the area surrounded by it represented the plastic strain energy. In further thermal cycling, the stress of the critical point was basically consistent and the plastic strain was accumulating at the same temperature in each cycle. After five cycles, the cumulative plastic strain of each cycle reached stability, and the value of the inelastic strain amplitude in one cycle after stabilization was 0.00456.

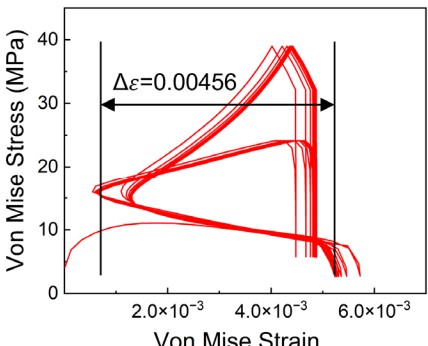

**Figure 5.** Stress-strain relationship of the critical point during ten thermal cycles.

Among many solder joint life prediction models, the Coffin–Manson equation with correction terms has a higher accuracy of low cycle fatigue life prediction, because it takes the conditions of thermal cycling into account such as the temperature amplitude and frequency. The modified Coffin–Manson model is the most widely-used model in the field of electronic packaging at present. The general expression of this model is [14,15]:

$$N_f = \frac{1}{2}\left(\frac{\Delta\gamma}{2\varepsilon_f}\right)^{\frac{1}{c}},\tag{1}$$

where $N_f$ is the number of cycles to fail; $\Delta\gamma$ is the range of equivalent inelastic shear strain; $\varepsilon_f$ is the fatigue ductility coefficient which is approximately equal to 0.325 for SnPb solder; $c$ is the fatigue ductility index, a constant related to the thermal cycle temperature, equal to 3.39 for this work. The range of equivalent inelastic strain can be extracted from the finite element simulation results, which has the following relationship with $\Delta\gamma$:

$$\Delta\gamma = \sqrt{3}\Delta\varepsilon,\tag{2}$$

where $\Delta\varepsilon$ is the range of equivalent inelastic strain.

The thermal cycling life of solder joint was calculated by taking $\Delta\varepsilon$ into the formulas above, and the calculated life was $5.2 \times 10^4$ cycles, which was not the typical low cycle fatigue failure. It could be considered that this optimized solder joint structure would not have fatigue failure cause by creep deformation during thermal cycling, and has a great thermal cycling reliability.

The cross-section of the solder joints after the thermal cycling test are shown in Figure 6. Within 1000 cycles, the solder joints still had no microcracks, voids, or other defects, which verified the simulation results and confirmed that the optimized structure of the new diamond chip resistor solder joint had good thermal cycling reliability. With the increase in temperature cycles, the eutectic phase of solder was only slightly coarsened, which would not significantly affect the mechanical properties of solder joints.

The Pb element in eutectic SnPb does not participate in the interface reaction. According to the Cu-Sn binary phase diagram, the IMCs can only be $Cu_6Sn_5$ and $Cu_3Sn$ below 350 °C. In addition, numerous studies have shown that during thermal cycling and aging, the reaction products of Cu-Sn interface are $Cu_3Sn$ near the side of Cu and $Cu_6Sn_5$ near the side of Sn [18–22]. Considering the significant difference in atomic ratios between the two compounds and the accuracy of EDS, we used EDS to determine the type of IMCs. The EDS results can be found in Figure S1 of the Supplementary Materials. The IMC of solder–pad interface near the Cu pad was continuous $Cu_3Sn$ where the Cu-Sn atomic ratio of EDS was very close to 3:1, while near the SnPb was continuous $Cu_6Sn_5$ with a microstructure of scallop where the atomic ratio was very close to 6:5. The IMC of solder–resistor interface was continuous layered $(Cu, Ni)_6Sn_5$ due to similar crystal structure of Cu and Ni [19].

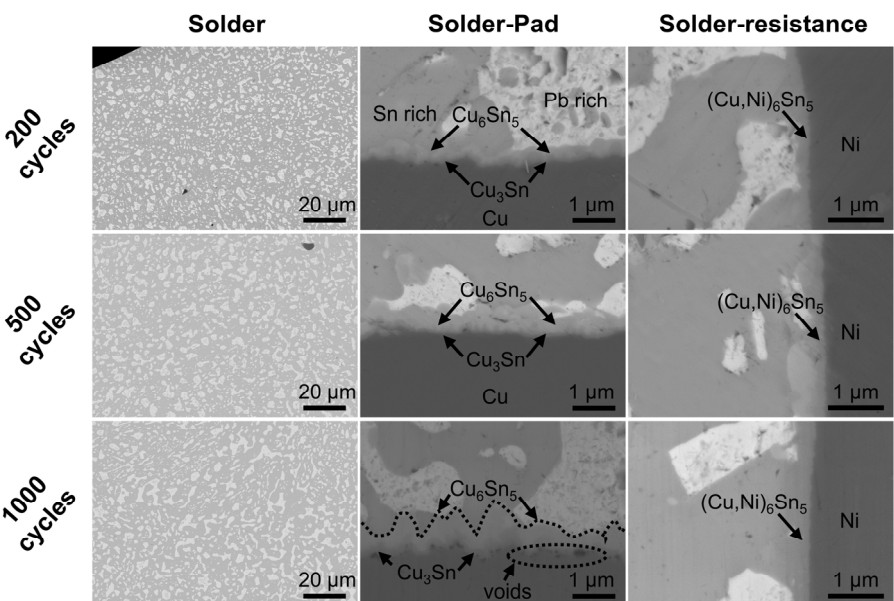

**Figure 6.** Cross-section SEM images of the solder joints after different thermal cycles.

As the thermal cycling test continued, the thickness of $Cu_6Sn_5$ and $Cu_3Sn$ at the solder–pad interface increased obviously. It was because high temperature promoted the diffusion of elements, and Cu atoms reacted with Sn atoms to generate $Cu_6Sn_5$, and $Cu_6Sn_5$ was converted into $Cu_3Sn$ gradually [18,19]. Kirkendall voids with diameter of 100–400 nm formed in the $Cu_3Sn$ layer and along the $Cu_3Sn/Cu$ interface. Those voids were generated because the diffusion rate of Cu in $Cu_3Sn$ was much higher than that of Sn, resulting in the accumulation of excess vacancies [20,21]. The interface strain caused by lattice mismatch or the elastic anisotropy between phases was another factor that led to the interface voids [22]. The thickness of IMC in the solder–resistor interface $(Cu, Ni)_6Sn_5$ is significantly lower than that of the solder–pad interface, which is due to the slow reaction of the main reaction elements Ni and Sn at the solder–resistor interface.

Current will cause the directional migration of atoms, which accelerates or inhibits the growth of IMCs, resulting in microcracks, voids, and other defects [23,24]. The mechanism of the influence of the current on the reliability is shown in Figure 7. A is the IMC of the solder–pad interface at the side where electrons flow out, B is the IMC of the solder–pad interface at the side where electrons flow in, C is the IMC of the resistor–solder interface at the side where electrons flow out, and D is the IMC of the resistor–solder interface at the side where electrons flow in. Because electrons are conducive to the migration of Cu atoms at B and Ni atoms at C, the IMCs at B and C are generally thicker than that at A and D.

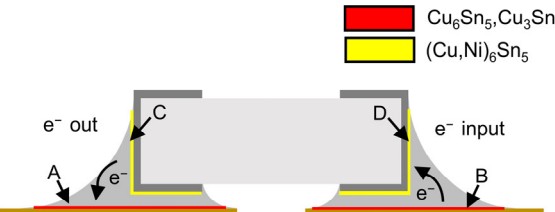

**Figure 7.** Diagram of the influence of current.

The section of the solder joints of the electrified samples were analyzed and there were no microcracks, voids, or other defects. The microstructure of the solder joints was basically the same as that of the untreated solder joints. In order to evaluate the effect of the current on the IMCs growth, the IMCs thickness under the different electrified conditions were calculated as shown in Table 1. The thickness of IMCs at A and D were basically the

same no matter whether the solder joint was electrified or not. Although the electron flow was not conducive to the migration of the main diffusion atoms here, the thermal effect generated by the current promotes the diffusion to a certain extent. The IMCs at B and C under electrified condition were thicker than that of the untreated solder joints. Within 1000 cycles, the difference of IMC thickness at B between the electrified and untreated solder joints was within 0.1 μm, while D was within 0.2 μm. It was proved that the solder joints' structure had excellent reliability against electromigration.

**Table 1.** The thickness of IMCs at different interfaces under thermal cycling.

| NO. of Cycles | Current | Thickness of IMCs (μm) | | | |
|---|---|---|---|---|---|
| | | A | B | C | D |
| 200 | off | 0.31 | - | 0.24 | - |
| 500 | off | 0.49 | - | 0.27 | - |
| 1000 | off | 0.72 | - | 0.32 | - |
| 200 | on | 0.30 | 0.34 | 0.35 | 0.27 |
| 500 | on | 0.51 | 0.59 | 0.44 | 0.31 |
| 1000 | on | 0.73 | 0.80 | 0.49 | 0.38 |

The sections of solder–pad interface with different aging time were shown in Figure 8. It can be seen that there were no cracks and delamination at the interconnection interface within 49 days of aging. The IMC of the unaged sample solder–pad interface was scallop-shaped $Cu_6Sn_5$, which was generated by the liquid phase reaction between the molten SnPb solder and the pad during reflow soldering. In the subsequent aging process, the reaction $Cu_6Sn_5 + 9Cu = 5Cu_3Sn$ occurred at the $Cu_6Sn_5$-Cu interface. The IMC thickness increased with the aging time. That was because the high temperature promoted atomic diffusion, and the growth of IMC was then dominated by atomic diffusion [25–29]. $Cu_6Sn_5$ grains ripen with the prolongation of aging time, large grains swallowed up small grains, grains became coarse, the number of grains decreased, and the IMC gradually became layered type [30–34].

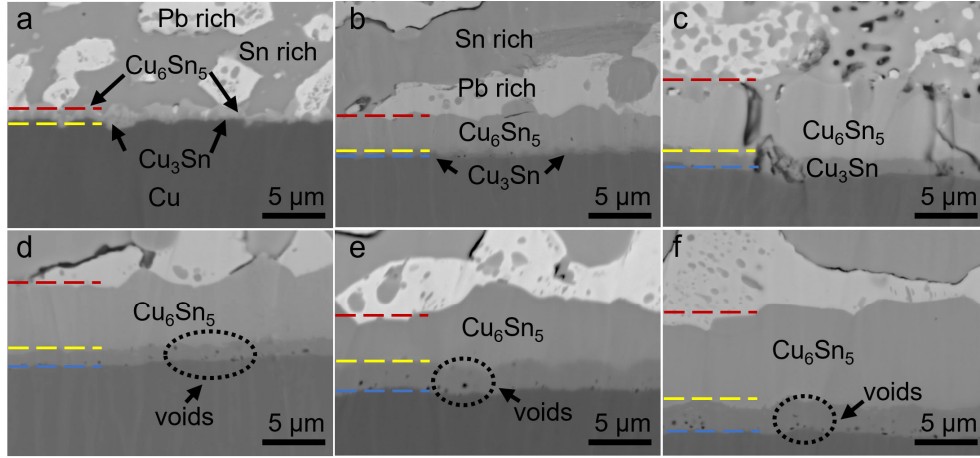

**Figure 8.** Cross-section SEM images of the solder–pad interface under the aging of (**a**) 0 day, (**b**) 4 days, (**c**) 16 days, (**d**) 25 days, (**e**) 36 days, (**f**) 49 days.

The relationship between IMCs thickness and aging time at the solder–pad interface is shown in Figure 9. The thickness of IMCs was proportional to the square root of aging time. The thickness of $Cu_6Sn_5$ and $Cu_3Sn$ at 150 °C met the following formulas:

$$h_{Cu_6Sn_5} = 0.994t^{0.5} + 0.453, \qquad (3)$$

$$h_{Cu_3Sn} = 0.4t^{0.5} + 0.1, \qquad (4)$$

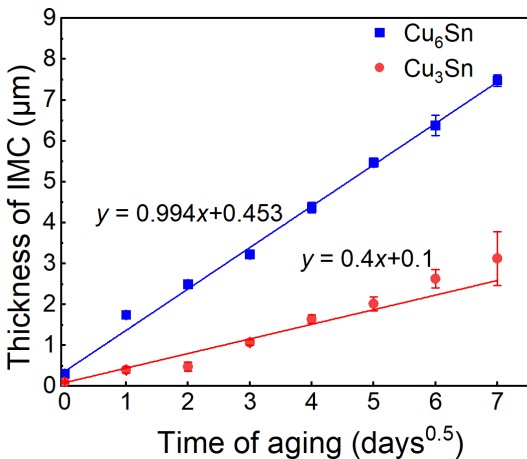

**Figure 9.** Curve fitting of IMC growth at solder–pad interface.

Through curve fitting, where $h$ is the thickness of IMC, $t$ is the time of aging. It was verified that IMC growth was dominated by element diffusion during aging and followed Fick's diffusion law. The IMC thickness at any temperature and time could be calculated by this fitting formula combining with the diffusion activation energy and the Arrhenius formula to evaluate the reliability of the solder joints at HTS.

Kirkendall voids appeared at the $Cu_3Sn$–Cu interface during aging. The atomic diffusion rate was higher than the temperature cycling, so Kirkendall voids were much more, but they did not gather to form cracks, which did not affect the reliability. Within 49 days of aging, there were no cracks, interface delamination, or other defects, and the eutectic phase of the solder was slightly coarsened with the aging time. The IMC at the solder–resistor interface was continuous layered $(Cu, Ni)_6Sn_5$, shown as Figure 10. At the beginning of aging process, the thickness of $(Cu, Ni)_6Sn_5$ increased, the Ni layer at the side of chip resistor was continuously consumed, and the Ni layer had completely reacted after 16 days of aging. At this time, the thickness of $(Cu, Ni)_6Sn_5$ did not increase significantly with the aging time, and the interface bonding was always tight, maintaining good reliability.

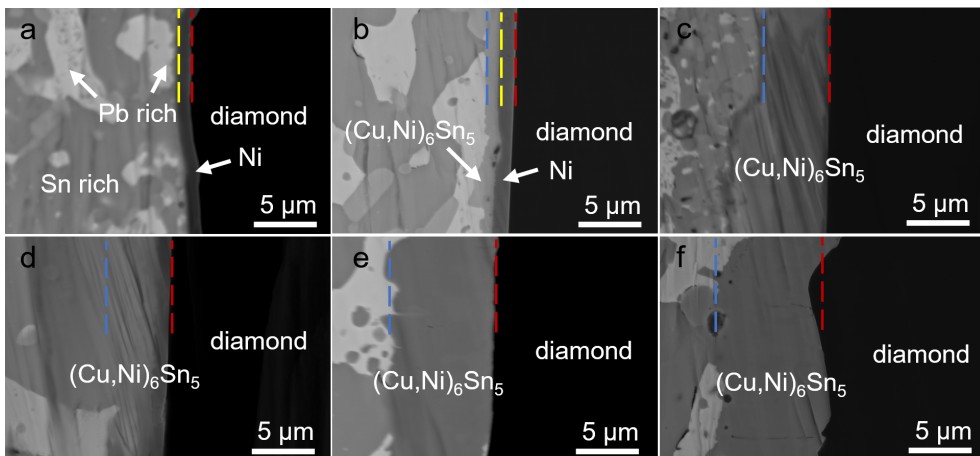

**Figure 10.** Cross-section SEM images of the solder–resistor interface under the aging of (**a**) 0 day, (**b**) 4 days, (**c**) 16 days, (**d**) 25 days, (**e**) 36 days, (**f**) 49 days.

## 4. Conclusions

This work systematically studied the reliability of the new chip diamond resistor solder joints, proposed a method to optimize the shape of the solder joints through numerical simulation, and determined the optimal solder volume of 0.05 mm³, which was verified to

be accurate by experiments. This optimized solder joint could guide the welding process of new chip resistors and improve the thermal cycling reliability of devices. The micromorphology change, IMCs growth, and evolution law of the new chip resistor solder joints under different loads were determined through a series of reliability tests. During the temperature cycling, the growth of IMCs was relatively slow, while under 2A current, the IMC growth was slightly promoted. The IMC thickness difference between the two sides of the chip resistor solder joint was within 0.2 μm, and IMCs grew faster during the aging process. The growth pattern of $Cu_6Sn_5$ and $Cu_3Sn$ were obtained by curve fitting, and the growth rate of IMCs at different temperatures could be predicted combined with the Arrhenius equation to evaluate the reliability. The results showed that the optimized solder joints had excellent thermal cycling reliability, anti-electric migration reliability, and HTS reliability, which could not only guide the welding process of new diamond chip resistors, but also accumulate some reliability data for it.

**Supplementary Materials:** The following supporting information can be downloaded at: https://www.mdpi.com/article/10.3390/coatings13040748/s1, Figure S1: EDS results of IMCs; Table S1: material properties at 25 °C; Table S2: Anand's constants of SnPb solder.

**Author Contributions:** Methodology, W.W. and X.S.; validation, G.L. and J.F.; formal analysis, G.L.; investigation, W.W. and G.L.; writing—original draft preparation, G.L.; writing—review and editing, S.W. and Y.W.; supervision, Y.T.; project administration, W.W. and Y.T.; funding acquisition, Y.T. All authors have read and agreed to the published version of the manuscript.

**Funding:** This research was supported by Heilongjiang Touyan Innovation Team Program (Grant No. HITTY-20190013) and National Natural Science Foundation of China (Grant No. U2241223).

**Institutional Review Board Statement:** Not applicable.

**Informed Consent Statement:** Not applicable.

**Data Availability Statement:** The datasets used and/or analysis results obtained in the current study are available from the corresponding author upon request.

**Conflicts of Interest:** The authors declare no conflict of interest.

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
