# Peer review of "Study on the Solder Joint Reliability of New Diamond Chip Resistors for Power Devices"

_coatings, doi:10.3390/coatings13040748_

Round 1

Reviewer 1 Report

The authors reported results on solder joint reliability used in new diamond chip resistors implemented in high-power devices with lent heat dissipation and high-frequency performance. Using numerical analysis and reliability experiments, the authors show that the optimized solder volume and the IMC thickness difference between the two sides of the chip resistor solder joint are 0.05 mm3 and  0.2 μm respectively. The authors show also that optimized solder joints had great anti-electromigration, temperature cycling and high-temperature storage reliability.

The results may be of interest to the readers of the journal, to improve the manuscript, the reviewer suggests that the authors take into account the comments below:

-          This chemical reaction, Cu6Sn5+Cu→Cu3Sn must be balanced.

-          The authors should be more specific in their captions, for example, they should mention the microscopy method used to obtain the images.

-          The EDS map will give more information about the chemical elements present in the different regions of their microscopic images. The stoichiometry mentioned for the different compounds should be discussed, it is not clear from the % atoms obtained from the EDS experiments.

Author Response

Thank you very much for your comments and suggestions on our work. We are very glad to hear your encouraging evaluation and so many helpful comments. We have carefully considered those advices and modified the manuscript. Revised portions are marked with yellow color in the manuscript. The detailed responses to the reviewer’s comments are listed in the appendix.

Reviewer 2 Report

After rigorous peer review, the paper is interesting, and it can be accepted for publication in this journal. However, a minor revision is needed. The questions and comments are appended below:

1.   Typos errors have been found throughout the paper; it should be double-checked.

2.   How do you know the IMCs solder-pad interface is Cu6Sn5 phase?

3.   In page 7, line 223, the sentence should be “The EDS results of IMC can be found in Fig. S1 of the supplementary material”.

4.   Scale bar and the values for solder in Fig. 6 should be black for clearer.

5.   The color of words in each figure for solder-pad and solder-resistance should also be black.

6.   Make larger the symbols and more thicker of error bars in Fig.9.

7.   The color of words in Fig.10(c and f) for (Cu,Ni)6Sn5 should be black.

Author Response

(The authors gave the same response as above.)

Reviewer 3 Report

Article no. 2314369

Title: Study on the Solder Joint Reliability of New Diamond Chip Resistor for Power Devices

Dear Editor,

The authors present a study on the optimization of the bonding process of the new resistors on a diamond substrate. The work is well documented and clearly presented. I recommend the publication of this work.

Best regards,

Author Response

Dear reviewer,

Thank you very much for your review and your affirmation of our article.

Best regards.

Reviewer 4 Report

In this work, the authors presented a study on the solder joint reliability of new diamond chip resistor for power devices applications. In my opinion, the work presented by authors here is great. The experiment and the simulation were conducted thoroughly as well as the authors provide deep discussion.
I have only minor comments:
1. Please improve the quality of the schematic (Fig.1-2)
2. Figure 9. The font is too small. Please increase it.

Author Response

(The authors gave the same response as above.)
